# The implementation of sex-and gender-based considerations in exercise-based randomized controlled trials in individuals with stroke: A cross-sectional study

Elise Wiley[1], Kenneth S. Noguchi[1], Hanna Fang[1], Kevin Moncion[1], Julie Richardson[1], Joy C. MacDermid[1,2], Ada Tang[1] *

**1** School of Rehabilitation Science, McMaster University, Hamilton, Ontario, Canada, **2** School of Physical Therapy, Western University, London, Ontario, Canada

* atang@mcmaster.ca

**Data Availability Statement:** All relevant data used to inform our analyses have been uploaded in our submission in supplementary file 1. Additionally, if

## Abstract

Emerging evidence suggests that sex-and gender-based factors may influence responses to exercise post-stroke. The Sex and Gender Equity in Research (SAGER) guidelines (2016) published international standards for terminology and considerations for research design and trial reporting. The extent to which sex- and gender-based considerations have been implemented in stroke exercise trials is currently unknown. The objective of this cross-sectional study was to compare the proportion of studies that have implemented sex/gender considerations before and after the publication of the SAGER guidelines. We conducted a comprehensive search of the literature to identify exercise-based trials in individuals with stroke. Study titles, abstracts, introductions (hypothesis statements), methods, results and discussions were assessed for adherence to the SAGER guidelines. The proportion of studies adhering to SAGER guidelines published prior to and including December 31, 2016 and from 2017-March 2023 were compared. Of the 245 studies identified, 150 were published before December 31, 2016, of which 0 (0%) titles/abstracts, 0 (0%) introductions, 21 (14.0%) methods, 8 (5.3%) results, and 7 (4.7%) discussion sections adhered to the SAGER guidelines, and 35 (23.3%) reported proper sex and gender terminology. Of the 95 studies published between 2017–2023, 0 (0%) title/abstracts, 1 (1.0%) introduction, 16 (16.8%) methods, 5 (5.3%) results, and 10 (10.5%) discussion sections adhered to the guidelines, and 37 (38.9%) of studies included proper terminology. The implementation of sex- and gender-based considerations in stroke exercise trials is low, but positively the reporting of proper terminology has increased since the publication of standardized reporting guidelines. This study serves as a call to action for stroke rehabilitation researchers to incorporate sex- and gender-based considerations in all stages of research studies, to improve the rigour and generalizability of findings, and promote health equity.

the manuscript is published, the data will be uploaded to the McMaster University dataverse (https://borealisdata.ca/dataverse/macstroke). The lead and corresponding authors have created a lab-specific public repository (MacStroke Canada Lab).

**Funding:** The author(s) received no specific funding for this work.

**Competing interests:** The authors have declared that no competing interests exist.

## Introduction

Stroke affects 100 million people globally [1]. It is the second leading cause of disability [2] that limits individuals' abilities to perform activities of daily living [3] and can contribute to a sedentary lifestyle [4]. There is strong systematic review evidence of physiological and psychosocial benefits of exercise after stroke [5], supporting engagement in regular exercise to lower risk for recurrent stroke and cardiovascular disease [6].

Emerging research suggests that sex- and gender-based factors may influence the effectiveness of exercise on health outcomes and participation behaviours after stroke [7–10]. While the terms sex and gender are interrelated, they are not interchangeable [11]. *Sex* refers to a biological construct, whereby an individual is characterized as male or female according to genetics, anatomy, and physiology [12]. Sex influences biological processes such as ageing and prevalence, diagnosis, severity, and outcomes of disease [13]. *Gender* refers to the social, environmental, cultural, and behavioural factors and choices that impact a person, whereby an individual is characterized as being a man, woman, girl, boy, or gender diverse [14]. Gender is fluid, multifaceted, complex, and encompasses constructs of gender roles, gender identity, gender relations and institutionalized gender [12].

Conflicting evidence of sex differences in outcomes following exercise after stroke have been previously reported. Females demonstrated greater improvement on tasks of selective attention and conflict resolution following 6 months of aerobic exercise [11)], and walking speed after 4 weeks of aerobic exercise [8]. Males experienced greater gains in strength and functional capacity (measured by the 6-minute walk test) in a 10-week unsupervised multi-component exercise intervention whereas females made greater improvements in a supervised environment [9], and a greater proportion of female participants dropped out of a 12-month cycling intervention after stroke compared to males [10]. Conversely, other studies have reported no sex differences in effects of exercise on a variety of cognitive and functional performance outcomes [15–19]. Although the intersection of sex and the role of social support in exercise environments and gender-related norms and responsibilities may have had important implications on the previous findings [9,10], no previous studies have explicitly sought to examine gender-based differences in the effects of exercise after stroke. Indeed, gender-related factors have influenced other areas of stroke research. Namely, women with stroke experience greater barriers to participating in stroke clinical trials due to competing caregiving roles [20], limited transportation and concerns with costs [21].

However, despite evidence of sex and/or gender differences in the effects of exercise [7–10] and females having poorer outcomes compared to males after stroke [22–25], sex- and gender-based considerations continue to be overlooked in all aspects of research [11,26]. Disaggregating results by sex and gender can provide insight on the importance of implementing interventions that are designed to target the specific needs or characteristics possessed by different subgroups of individuals [27], and can also be used to inform sample size calculations and facilitate meta-analyses [28]. However, it is also well known that females are under-represented in clinical research [29]: a review of 30 Cochrane systematic reviews comprising of 258 cardiovascular treatment trials and 286,302 participants reported that only 27% of participants were of female sex [30]. Moreover, in 2016, an audit of a random sample of 57 randomized-controlled trials published in high-impact journals reported that only 20% of the included studies disaggregated the results by sex, and none conducted gender-based analyses [31]. Given these notable gaps in research, guidelines for reporting Sex and Gender Equity in Research (SAGER) were published in 2016 to enhance reporting of sex-and gender-based considerations in scientific research [11].

Several studies have examined the extent to which sex and gender have been considered in research areas such as shared-decision making [32], military veterans with chronic pain [33], acute stroke therapies [34], sport and exericse research in a general adult population [35]. To date, no studies have examined the extent to which sex- and gender-based considerations have been incorporated in exercise-based randomized controlled trials in individuals with stroke. Thus, the objective of this study was to compare the proportion of studies that have implemented sex- and gender-based considerations in exercise-based randomized controlled trials in individuals with stroke before and after the publication of the SAGER guidelines.

## Methods

### Study design

This was a cross-sectional study of exercise-based randomized controlled trials in individuals with stroke. This type of design has been previously used in studies with similar scope of research questions [36,37]. Where applicable, the STROBE cross-sectional checklist was used when writing our results [38]. This study did not involve primary data collection on human participants, and thus consent to participate was not relevant.

A comprehensive search was conducted in five electronic databases (Embase, Medline, AMED, CINAHL and PsycINFO) from inception to March 22, 2023. We also examined reference lists of previous systematic reviews to capture any additional studies. To ensure that all relevant studies were captured, titles and abstracts were screened by two independent reviewers (EW and KSN, HF or KM). A third reviewer (AT) was consulted to resolve any discrepancies encountered among the two reviewers.

Studies were included if they met the following inclusion criteria:

*Participants*: The population of interest was adults ($\geq$18 of age) post-stroke. No restrictions were placed on time since stroke, or number or types of strokes. Participants could be community dwelling or undergoing inpatient rehabilitation.

*Study Design*: Randomized controlled trials, including randomized cross-over studies and secondary analyses of trials.

*Intervention*: Aerobic exercise (including aquatic), traditional resistance training, functional, or mixed/multicomponent training (e.g., aerobic exercise + resistance training) interventions of any length, duration, and frequency. Exercise interventions that included an additional balance, stretching/flexibility, conventional/usual therapy or non-exercise (e.g., cognitive training or education) component were also eligible.

*Comparator*: There were no restrictions on the type of comparator group and could include a form of exercise program (e.g., low-intensity aerobic or balance training), usual care/conventional therapy, wait list, or no intervention.

*Outcomes*: Any physiological, psychosocial, or functional outcomes.

*Language*: Studies published in English.

To narrow the focus of this study, we excluded studies that incorporated exergames, functional electrical stimulation, transcranial magnetic stimulation, transcranial electrical nerve stimulation (TENS), robotics, exoskeletons, mirror therapy, or virtual reality technologies into the intervention (i.e., 3-dimensional devices), trunk, or balance and mobility-focused interventions, such as bridging, yoga, Tai Chi, and gait training (e.g., low intensity body-weight supported treadmill training or uneven ground training stepping with perturbations). We only included interventions where the primary exercise focus could be classified as cardiorespiratory/aerobic and/or muscular strength training (includes functional training) per the American College of Sports Medicine guidelines [39]. We did not include thesis dissertations, commentaries, nor poster abstracts.

## Statistical analyses

Included studies were evaluated by one reviewer (EW) with extensive knowledge in sex and gender considerations. Data related to study demographic information including the total sample size, proportion of female participants (6 categories: $\leq$ 20, 21–40, 41–59, 60–79, 80–99, $\geq$100), the country in which the study was conducted, and the type of exercise intervention (5 categories: aerobic, functional, multicomponent, strength, > 2 intervention groups) were extracted for the purpose of characterizing the studies.

Table 1 provides the SAGER guidelines [11] that were referenced to determine the implementation of sex and gender in research studies. Studies that fulfilled the outlined criterion within each section of the SAGER guidelines were coded as "YES", while studies that failed to address each specific criterion were coded as "NO". The number of "YES" codes were tallied for each section. Titles and abstracts were coded as "NA" (not applicable) if the study was conducted in both male and female participants, as the SAGER criterion for title and abstract reporting would not be applicable in such circumstance. Importantly, to fully explore the scope of sex-based considerations in exercise trials among individuals with stroke, we also considered studies that reported on sex-specific reference values, main effect of sex, subgroup or disaggregated sex-based analyses as fulfilling the criteria for results section.

Information regarding the use of proper sex and gender terminology throughout the manuscript were also extracted and evaluated. Studies were assigned a code of "YES" if the correct terminology for sex and gender were consistently reported (e.g., studies that used the

**Table 1. Criteria for the implementation of sex and gender considerations in research studies established in the Sex and Gender Equity in Research (SAGER) guidelines.**

| Recommendations per section of the article | |
|---|---|
| Title and Abstract | If only one sex is included in the study, or if the results of the study are applied to only one sex or gender, the title and abstract should specify the sex and gender of the participants. |
| Introduction | Authors should report, where relevant, whether sex and/or gender differences may be expected. |
| Methods | Authors should report how sex and gender were taken into account in the design of the study, whether they ensured adequate representation of males and females, and justify the reasons for any exclusion of males or females. |
| Results | When appropriate, data should be routinely presented disaggregated by sex and gender. Sex-and gender-based analyses should be reported regardless of positive or negative outcomes. In clinical trials, data on withdrawals and dropouts should also be reported disaggregated by sex. |
| Discussion | The potential implications of sex and gender on the study results and analyses should be discussed. If a sex and gender analysis was not conducted, the rationale should be given. Authors should further discuss the implications of the lack of such analysis on the interpretation of the results. |
| Sex and Gender Terminology (Adapted from SAGER guidelines) | Authors should use the terms sex and gender carefully to avoid confusing both terms. When referring to biological sex, authors should use terminology males and females rather than men and women throughout the manuscript, and this terminology should not be used interchangeably. If authors report the gender of participants, appropriate terminology related to the construct of gender should be used (e.g., gender identity: men, women, gender diverse, prefer to self-describe). |

terminology "sex" and referred to participants as "males & females") throughout the entirety of the manuscript. The code "NO" was assigned if the study failed to use any sex or gender terminology correctly (e.g., used term "gender" when referring to sex and also used term "men & women" when reporting on sex of participants in different sections of the manuscript, including when reporting findings of previous studies in the discussion section). The terms "men and women" should be used in the context of gender-based considerations. Finally, the code "NR" (i.e., not reported) was assigned to studies that failed to include any sex and/or gender terminology.

We compared the proportion of studies that included SAGER recommendations between studies published prior to and including December 31, 2016 (indicative of year that guidelines were published) and from January 1, 2017- March 22, 2023, after the guidelines were released. Frequencies and percentages were used to describe categorical data. The number of studies including sex and/or gender considerations (0 = No, 1 = Yes, 2 = Not Reported) were tabulated for each section, including the abstract, introduction, methods, results, discussion and use of appropriate sex and gender terminology. Pearson's Chi-squared tests were conducted for each section to explore whether statistically significant differences existed in the observed frequencies of "YES" (i.e., adhered to SAGER guidelines) values between the two timepoints. All statistical analyses were performed on Stata/IC 15.1 (StataCorp, College Station, TX, USA).

## Results

Fig 1 provides a flow chart of studies included in this analysis. There were 245 studies included in our cross-sectional analysis, of which 150 studies were conducted prior to and including December 31, 2016 and 95 studies were conducted between January 1, 2017 to March 22, 2023. A complete list of included references can be seen in the supporting information file (S1 Table).

Table 2 provides a summary of study characteristics. Approximately 42% of studies were aerobic-based exercise interventions. One-third of studies (34.3%) had sample sizes of 21–40 participants, while approximately 10% of studies had sample sizes greater than 100 participants. Studies were conducted across 30 countries, most commonly in the United States of America (n = 46; 18.8%), Canada (n = 30; 12.2%), South Korea (n = 22; 9.0%), Australia (n = 16; 6.5%), Brazil (n = 15; 6.1%), and China (n = 12; 4.9%), United Kingdom (n = 11; 4.5%), Taiwan (n = 10; 4.1%) and Sweden (n = 11; 4.5%). For studies published up to December 31, 2016, sample sizes ranged from 7 [40] to 408 [41] participants, with proportion of female participants (i.e., sex, number (%) of females [42]) ranging from 0% [43,44] to 65.0% [45]. For studies published between January 1, 2017 and March 22, 2023, sample sizes ranged from 7 [46,47] to 347 [48] participants, with proportion of female participants ranging from 6.7% [49] to 83.3% [50].

Table 3 presents the implementation of sex- and gender-based considerations in exercise-based randomization controlled in each section of the research article any year prior to and including December 31, 2016 and January 1, 2017- March 2023 (following the publication of SAGER guidelines), as well as how sex and/or gender considerations were reported. The percentage of studies including sex- and gender-based considerations increased slightly in each section following the publication of the SAGER guidelines, though not significantly. Encouragingly, however, appropriate reporting of sex and gender terminology increased by 15.6%, from 23.3% to 38.9% (p = 0.01). We note that all considerations were related to biological sex, whereas gender-based factors were not considered in the study design, results, and interpretation of findings. Table 4 provides the proportion of studies incorporating sex- and gender-based considerations in exercise-based RCTs in individuals with stroke prior to and including the December 31, 2016 and between January 1, 2017- March 22, 2023.

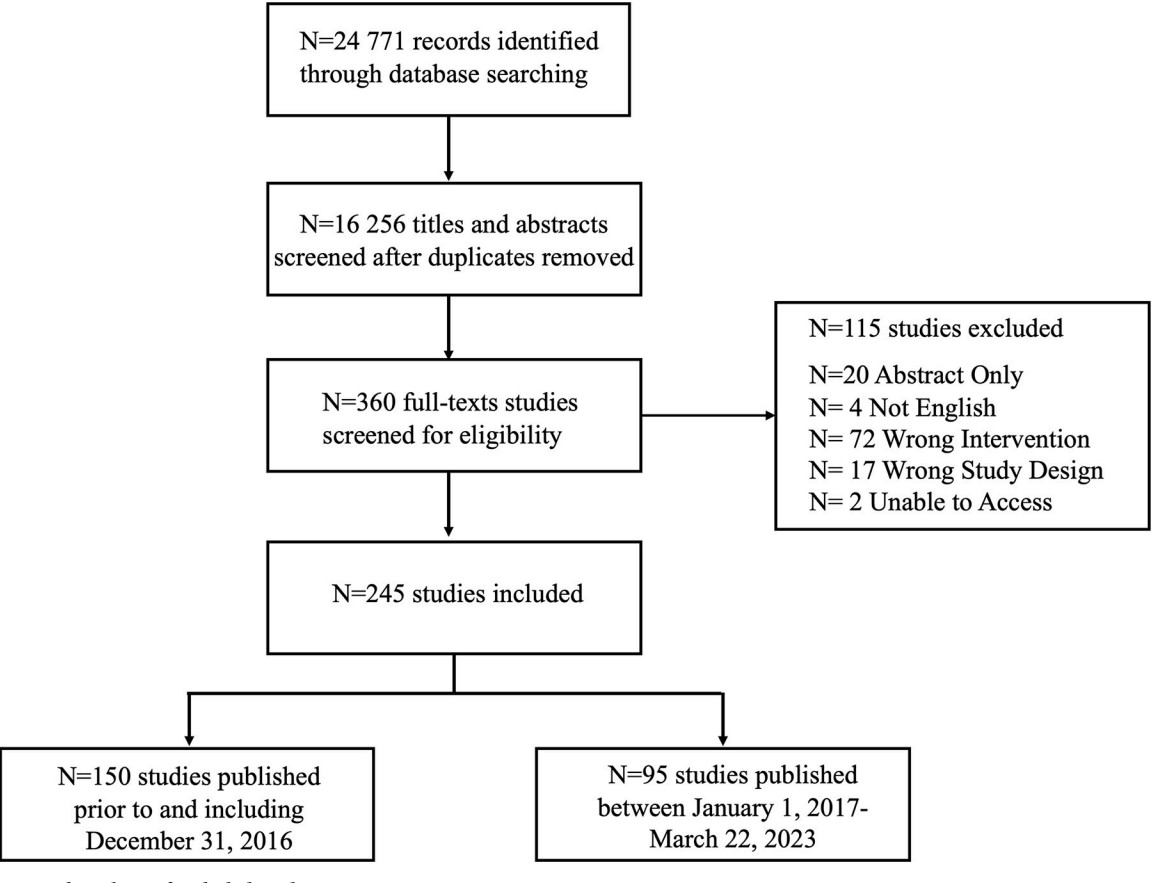

**Fig 1. Flow chart of included studies.**

## Discussion

Incorporating sex- and gender-based considerations into health research design, interpretation and study reporting is essential for moving towards evidence that is equitable, inclusive and representative of populations to inform best practices. Indeed, exercise is crucial for secondary prevention of stroke [6]. This study evaluated exercise trials for stroke, including aerobic exercise, resistance, functional, and multicomponent training, to determine the extent to which studies addressed recommendations provided in the SAGER guidelines [11] before and after its publication. As the body of exercise literature in stroke continues to grow, it is important to consider how sex- and gender-based considerations can be incorporated.

With the exception of one study published in 2019 [7], our analysis revealed that the majority of articles did not adequately implement SAGER guidance in the introduction/ background sections, and hypothesis statements of the remaining studies did not describe potential or expected sex and/or gender differences in their findings. Including an *a priori* hypothesis specifying the expected sex differences minimizes risks of post-hoc fishing and data-derived hypothesis testing [27]. As such, per SAGER recommendations, study results should be disaggregated by sex where possible, and of equal importance that the rationale for these analyses are provided in earlier sections of the manuscript.

Most of the studies that satisfied study design recommendations outlined by the SAGER guidelines [11] specified sex as a stratification variable in the randomization procedure or as a covariate in statistical models; interestingly, however, female participants represented less than

**Table 2. Study demographic information (n = 245 studies).**

| Study sample size, n (%) | |
|---|---|
| ≤ 20 | 48 (19.6%) |
| 21–40 | 84 (34.3%) |
| 41–59 | 44 (17.9%) |
| 60–79 | 33 (13.5%) |
| 80–99 | 12 (4.9%) |
| ≥100 | 24 (9.8%) |
| **Proportion of females, n (%)** | |
| ≤25% | 39 (15.9%) |
| 26–49% | 157 (64.1%) |
| 50–74% | 32 (13.1%) |
| 75–100% | 1 (0.4%) |
| Not Reported | 16 (6.5%) |
| **Intervention type, n (%)** | |
| Aerobic | 103 (42.0%) |
| Functional | 52 (21.2%) |
| Multicomponent/mixed | 40 (16.3%) |
| Strength | 40 (16.3%) |
| >2 Intervention Groups (e.g., aerobic or strength group) | 10 (4.1%) |
| **Continent, n (%)** | |
| North America | 77 (31.4%) |
| Asia | 68 (27.8%) |
| Europe | 60 (24.5%) |
| Oceania | 22 (9.0%) |
| South America | 15 (6.1%) |
| Africa | 3 (1.2%) |

50% of the samples in the majority of studies. We acknowledge that both sex- and gender-based challenges exist to enrolling representative proportions of females/women into clinical research studies such as strict eligibility criteria (e.g., enrolling males only; low maximum age limit where females, who tend to live longer, will exceed the age limit) and competing gender roles and responsibilities [21]. Thus, it is important to implement strategies to increase the proportion of females/women with stroke recruited to and retained in exercise-based clinical studies.

We also noted that two studies [79,80] included insight on age- and sex-specific reference values in the study design and results sections, respectively. Indeed, sex-specific reference values are important to consider, since anatomical and physiological factors may greatly impact an individual's response to an outcome [90]. We acknowledge that many outcome measures lack data on sex-specific reference values [90], and thus further measurement studies are highly warranted to eventually allow for the standard reporting practice of presenting outcome data by sex- and gender-specific reference values.

Our finding of ~5% of studies presenting sex-related findings in the results section is low, but in line with a previous cross-sectional analysis study examining the implementation of sex-and gender-based considerations in health-related Canadian RCTs (e.g., pharmacological, surgical, rehabilitation interventions, etc.). This study also reported that 5% of the included studies performed sex-based subgroup analyses [36]. Thus, it is clear that the lack of analyses and reporting on sex and/or gender-related findings is consistent in health research. As we previously alluded to, the under-representation of female participants in the included studies also impose power concerns for performing sex-based analyses. An additional common

**Table 3. Sex-and gender-based considerations in exercise-based RCTs in individuals with stroke.**

| Section of Article | Prior to and including December 1, 2016 N Total studies (%) | N studies sex-based | N studies gender-based | January 1, 2017-March 22, 2023 N studies (%) | N studies sex-based | N studies gender-based |
|---|---|---|---|---|---|---|
| **Abstract*** | **0 (0%)** | **NA** | **NA** | **0 (0%)** | **NA** | **NA** |
| **Introduction** | **0 (0%)** | **NA** | **NA** | **1 (1.1%)** [7] | **1 (1.1%)** [7] | **0 (0%)** |
| **Study Design** | **21 (14.0%)**** | **21 (14.0%)** | **0 (0%)** | **16 (16.8%)**** | **16 (16.8%)** | **0 (0%)** |
| Sex and or gender-based recruitment considerations | 0 (0%) | NA | NA | 0 (0%) | NA | NA |
| Stratification by sex and/or gender in randomization | 9 (6.0%) [16,51–58], | 9 (6.0%) [16,51–58] | 0 (0%) | 2 (2.1%) [59,60] | 2 (2.1%) [59,60] | 0 (0%) |
| Covariate and/or confounder variable of sex and/or gender | 10 (6.7%) [10,53,61–68] | 10 (6.7%) [10,53,61–68] | 0 (0%) | 10 (10.5%) [8,69–77] | 10 (10.5%) [8,69–77] | 0 (0%) |
| Intentions to report findings by sex and/or gender (i.e., subgroup analysis, interaction terms) | 4 (2.7%) [9,10,17,18] | 4 (2.7%) [9,10,17,18] | 0 (0%) | 2 (2.1%) [7,8] | 2 (2.1%) [7,8)] | 0 (0%) |
| Intentions to perform sex and/or gender-based sensitivity analyses<br>Intentions to report sex-specific reference values for outcome measures | 0 (0%)<br>0 (0%) | NA<br>0 (0%) | NA<br>NA | 1 (1.3%) [78]<br>2 (2.1%) [79,80] | 1 (1.3%) [78]<br>2 (2.1%) [79,80] | 0 (0%)<br>0 (0%) |
| **Results**<br>Main effect of sex and/or gender reported<br>Interaction term in models or subgroup analysis (i.e., disaggregate by sex and /or gender)<br>Values on sex-specific reference values for outcomes reported | **8 (5.3%)**<br>3 (2.0%) [63,64,66]<br>5 (3.3%) [9,10,16–18]<br>0 (0%) | **8 (5.3%)**<br>3 (2.0%) [63,64,66]<br>5 (3.3%) [9,10,16–18]<br>0 (0%) | **0 (0%)**<br>0 (0%)<br>0 (0%)<br>NA | **5 (5.3%)**<br>1 (1.1%) [71]<br>2 (2.1%) [7,8]<br>2 (2.1%) [79,80] | **5 (5.3%)**<br>1 (1.1%) [71]<br>2 (2.1%) [7,8]<br>2 (2.1%) [79,80] | **0 (0%)**<br>0 (0%)<br>0 (0%)<br>0 (0%) |
| **Discussion** | **7 (4.7%)** | **7 (4.7%)** | **0 (0%)** | **10 (10.5%)** | **10 (10.5%)** | **0 (0%)** |
| Implications of sex and/or gender on findings | 2 (1.3%) [10,17] | 2 (1.3%) [10,17]<br>Note. 2 studies only briefly acknowledged implications of sex. | 0 (0%) | 3 (3.2%) [7,81,82] | 3 (3.2%) [7,81,82]<br>Note. 2 studies only briefly acknowledged implications of sex. | 0 (%) |
| Acknowledgement of further sex and gender-based research and/or targeted sex and gender-based exercise programs | 1 (0.67%) [9] | 1 (0.67%) [9] | 0 (0%) | 2 (2.1%) [8,77] | 2 (2.1%) [8,77] | 1 (1.1%) [7] |

(*Continued*)

**Table 3.** (Continued)

| Section of Article | Prior to and including December 1, 2016 N Total studies (%) | N studies sex-based | N studies gender-based | January 1, 2017-March 22, 2023 N studies (%) | N studies sex-based | N studies gender-based |
|---|---|---|---|---|---|---|
| Limitations (i.e., unequally proportion of males and females, difficulties with recruitment and/or adherence of specific sex and/or gender | 4 (2.7%) [83–86] | 4 (2.7%) [83–86] | 0 (0%) | 5 (5.3%) [46,47,77,87,88] 1 (1.1%) [89] study reported equal representation of sexes between intervention groups as strength. | 5 (5.3%) [46,47,77,87,88] 1 (1.1%) [89] equal proportion of males and females a strength of study. | 0 (0%) |
| **Proper Use of Sex and Gender Terminology** *** | **35 (23.3%)** | **35 (23.3%)** | **0 (0%)** | **37 (38.9%)** | **37 (38.9%)** | **1 (1.1%)** [7] (included both proper sex and gender terminology. Study counted once in report of total studies. |

*The majority of Pre-SAGER studies (n = 148; 98.7%) were conducted in both male and female participants, and thus the code "NA" was assigned. In the remaining 2 studies, (1.3%) sex of participants were not identified and were assigned "NO". All 95 Post-SAGER were conducted in both male and female sex, and thus were assigned a code of "NA" based on the SAGER recommendations for the title and abstract section.

**Some studies included multiple sex and/or gender considerations per section, but were counted once in report of total studies.

***See supplementary document for studies that implemented proper and consistent use of sex and gender terminology throughout manuscript.

finding between this current study and previous work [32,37] is the non-representation of non-binary individuals and other analyses associated with any of the four constructs of gender. Gender-based analyses are particularly needed to identify inequalities between men, women, and gender-diverse individuals in health and health care [91]. It is encouraging, however, to note the recent development and validation of efficient, yet comprehensive gender indices that can be administered by researchers to capture different constructs of gender [92,93], and can facilitate and increase the proportion of gender-based analyses.

With respect to the discussion sections of the included trials, less than 10% of studies (17/245 studies) met reporting standards. In most cases, these studies acknowledged the misbalanced representation of male and female participants as a limitation, which is indeed good practice to report on, according to the SAGER guidelines [11]. However, out of the seven studies that included an interaction term in models or conducted subgroup sex-based analyses,

**Table 4.** Sex and gender reporting of exercise-based randomized controlled trials in people with stroke.

| Criteria | Pre-SAGER (n = 150) | Post-SAGER (n = 95) | p-value |
|---|---|---|---|
| Title/Abstract | 0 (0%)[a] | 0 (0%)[a] | >0.99 |
| Introduction | 0 (0%) | 1 (1.0%) | NS[b] |
| Methods | 21 (14.0%) | 16 (16.8%) | 0.55 |
| Results | 8 (5.3%) | 5 (5.3%) | 0.98 |
| Discussion | 7 (4.7%) | 10 (10.5%) | 0.08 |
| Terminology | 35 (23.3%) | 37 (38.9%) | **0.01*** |

*Note.* NS = Not Significant.

*Reflects $p<0.05$.

[a]Refer to Table 1 for criteria for Titles/Abstracts

[b]Specific p-value not provided as considered a rare event (<5 values).

only one study [7] provided a thorough discussion on physiological considerations related to the findings. Future studies should be mindful to elaborate on the physiological or social-cultural factors contributing to the outcomes, especially when those analyses were conducted.

Finally, in terms of use of sex and gender terminology, approximately 25% and 40% of studies conducted prior to and including 2016 and between 2017–2023 incorporated the appropriate terminology, respectively. Our findings corroborate previous reports of sex and gender terminology being commonly used interchangeably in research studies [32]. Specifically, the authors reported that 48% of shared decision-making interventions used appropriate sex-based terminology. We most commonly noted that authors lacked consistency in their selection of terminology throughout different sections of the study. A potential solution to overcoming challenges around consistency includes the incorporation of a statement explaining the choice of terminology. For example, previous studies have stated that terms "men" and "women" would be used throughout the study to acknowledge and reflect the intersectionality between sex and gender factors, even though the aim of the study was to explore sex-based differences [25,94].

Despite concerns of sex- and gender-based considerations often being overlooked in the design, study implementation, and scientific reporting in research studies [11], our findings demonstrate that some progress is being made towards the increased implementation of proper use of sex and gender terminology in exercise-trials in individuals in stroke. These increases may be attributed to factors such as the publication of the SAGER guidelines and the initiatives taken by major funding agencies. Interestingly, the greatest proportion of exercise-based trials conducted among individuals with stroke that incorporated sex and/or gender-based considerations were conducted in Canada (over half of Canadian trials fulfilled one or more criteria), which may be in part due to the leading initiatives taken by major federal funding agencies and institutions, such as the Canada Tri-Council Agencies to mandate the inclusion of sex and gender in research grants [95]. As the development of tools and guidelines to assess and report on sex and gender constructs continue to evolve, we encourage researchers around the world to consider how both sex and gender may be adequately incorporated into trials.

## Limitations

We acknowledge that this study has limitations. Firstly, we were limited to only including randomized controlled trials in our study. It is of equal importance to examine the extent to which sex-and gender-based considerations are incorporated in various study designs, such as longitudinal cohort, cross-sectional studies or qualitative research studies, however, including other study designs in addition to randomized controlled trials would not have been feasible due to the extensive number of studies conducted to date. We also excluded technology-based, low intensity balance and mobility interventions as we did not have capacity to include such a broad range of studies. Thus, future studies should consider examining the implementation of sex-and gender-based considerations in other research methodologies and designs, as well as encompass mobility and balance-based exercise studies in individuals with stroke.

## Conclusion

This study was the first to examine the extent to which sex-and gender-based considerations are implemented in aerobic, resistance, and multicomponent training interventions in individuals with stroke. Adherence to each section of the internationally renowned SAGER guidelines were examined. Of the 245 studies included in this study, our findings indicated a staggeringly low proportion of studies that incorporated sex- and-gender-based considerations in the title

and/or abstract, introduction, methods, results, and discussion sections. Moreover, we also noted frequent usage of improper sex and gender terminology, though has decreased within the past seven years. Our study ultimately serves as a call to action for stroke rehabilitation researchers to incorporate sex- and gender-based considerations in all stages of research studies, to improve the rigour and generalizability of findings, and promote health equity.

## Supporting information

**S1 Table. Sex and gender reporting in included references.**
(DOCX)

## Author Contributions

**Conceptualization:** Elise Wiley, Kenneth S. Noguchi, Julie Richardson, Ada Tang.

**Data curation:** Kenneth S. Noguchi, Hanna Fang, Kevin Moncion.

**Formal analysis:** Elise Wiley.

**Methodology:** Elise Wiley, Kenneth S. Noguchi.

**Software:** Elise Wiley.

**Writing – original draft:** Elise Wiley.

**Writing – review & editing:** Kenneth S. Noguchi, Hanna Fang, Kevin Moncion, Julie Richardson, Joy C. MacDermid, Ada Tang.

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
