## [Decision Letter · Decision Letter 0]

12 Jul 2024

PONE-D-24-13337The Implementation of Sex-and Gender-Based Considerations in Exercise-Based Randomized Controlled Trials in Individuals with StrokePLOS ONE

Dear Dr. Tang,

Thank you for submitting your manuscript to PLOS ONE. After careful consideration, we feel that it has merit but does not fully meet PLOS ONE’s publication criteria as it currently stands. Therefore, we invite you to submit a revised version of the manuscript that addresses the points raised during the review process. I have attached the comments of reviewer 2.

We look forward to receiving your revised manuscript.

Kind regards,

Felicianus Anthony Pereira, MS Advanced Physiotherapy

Guest Editor

PLOS ONE

Reviewers' comments:

Reviewer's Responses to Questions

**Comments to the Author**

1. Is the manuscript technically sound, and do the data support the conclusions?

Reviewer #1: Yes

Reviewer #2: Yes

2. Has the statistical analysis been performed appropriately and rigorously? 

Reviewer #1: Yes

Reviewer #2: Yes

3. Have the authors made all data underlying the findings in their manuscript fully available?

Reviewer #1: Yes

Reviewer #2: Yes

4. Is the manuscript presented in an intelligible fashion and written in standard English?

Reviewer #1: Yes

Reviewer #2: Yes

5. Review Comments to the Author

Reviewer #1: The authors showed great understanding of the subject matter. This study could help in the discussion of studies in order to situate them in the right perspectives. It will also help to use the right terminologies in studies.

Reviewer #2: The Implementation of Sex-and Gender-Based Considerations in Exercise -Based Randomized Controlled Trials in Individuals with Stroke.

Excellent work!

1. Kindly rephrase line 110 to 114 or simplify the sentence(s) for easy comprehension.

2. Was there any specific reason(s) for the non-inclusion of thesis dissertations?

6. PLOS authors have the option to publish the peer review history of their article (what does this mean?). If published, this will include your full peer review and any attached files.

Reviewer #1: No

Reviewer #2: **Yes:**

---

## [Author Response · Author response to Decision Letter 0]

19 Jul 2024

Response Document: “The Implementation of Sex-and Gender-Based Considerations in Exercise-Based Randomized Controlled Trials in Individuals with Stroke: A Cross-Sectional Study”

We thank the reviewers for their time and positive reviews of our study, and for their appreciation of the importance of incorporating sex and gender considerations into research. We have provided a point-by-point response to Reviewer 2’s comments below, and have uploaded a clean and revised manuscript to the portal. Changes to the manuscript have been added in yellow highlight. 

Reviewer 2

Excellent work!

1. Kindly rephrase line 110 to 114 or simplify the sentence(s) for easy comprehension.

AUTHOR RESPONSE: We thank the reviewer for this suggestion around simplifying the sentence in lines 110-114. The revised wording is provided below. 

Lines 110-114: However, despite evidence of sex and/or gender differences in the effects of exercise [11-14] and females having poorer outcomes compared to males after stroke [22–25], sex- and gender-based considerations continue to be overlooked in all aspects of research [7,26]. 

2. Was there any specific reason(s) for the non-inclusion of thesis dissertations?

AUTHOR RESPONSE: We thank the reviewer for this comment. There are a few reasons that thesis dissertations were excluded: 1) We opted to only include studies that were peer reviewed to provide an appraisal of the literature meeting standards of rigour, quality and ethical conduct held by academic journals; 2) The individual studies included in thesis dissertations may have in fact been previously published in peer-reviewed journals, and thus were excluded to avoid the risk of duplication; and 3) Different institutions have varying requirements for the format of thesis dissertations, and thus the appraisal of content within individual dissertations would have been challenging to report.

---

## [Editor Report · Decision Letter 1]

25 Jul 2024

The Implementation of Sex-and Gender-Based Considerations in Exercise-Based Randomized Controlled Trials in Individuals with Stroke: A Cross-Sectional Study

PONE-D-24-13337R1

Dear Dr. Tang,

We’re pleased to inform you that your manuscript has been judged scientifically suitable for publication and will be formally accepted for publication once it meets all outstanding technical requirements.

Kind regards,

Felicianus Anthony Pereira, MS Advanced Physiotherapy

Guest Editor

PLOS ONE

---

## [Editor Report · Acceptance letter]

16 Sep 2024

PONE-D-24-13337R1 

PLOS ONE

Dear Dr. Tang, 

I'm pleased to inform you that your manuscript has been deemed suitable for publication in PLOS ONE. Congratulations! Your manuscript is now being handed over to our production team.

Kind regards, 

on behalf of

Dr. Felicianus Anthony Pereira 

Guest Editor

PLOS ONE